# Dual Optoelectronic Organic Field-Effect Device: Combination of Electroluminescence and Photosensitivity

**DOI:** 10.3390/molecules29112533

**Published:** 2024-05-28

**Authors:** Vasiliy A. Trukhanov, Andrey Y. Sosorev, Dmitry I. Dominskiy, Roman S. Fedorenko, Victor A. Tafeenko, Oleg V. Borshchev, Sergey A. Ponomarenko, Dmitry Y. Paraschuk

**Affiliations:** 1Faculty of Physics, Lomonosov Moscow State University, Leninskie Gory 1/62, Moscow 119991, Russia; 2Enikolopov Institute of Synthetic Polymeric Materials, Russian Academy of Science, Profsoyuznaya 70, Moscow 117393, Russia; 3Department of Chemistry, Lomonosov Moscow State University, Leninskie Gory 1/3, Moscow 119991, Russia

**Keywords:** organic field-effect transistors, light-emitting transistors, electroluminescence, organic phototransistors, organic semiconductors, thiophene-phenylene co-oligomers, density functional theory, charge transport

## Abstract

Merging the functionality of an organic field-effect transistor (OFET) with either a light emission or a photoelectric effect can increase the efficiency of displays or photosensing devices. In this work, we show that an organic semiconductor enables a multifunctional OFET combining electroluminescence (EL) and a photoelectric effect. Specifically, our computational and experimental investigations of a six-ring thiophene-phenylene co-oligomer (TPCO) revealed that this material is promising for OFETs, light-emitting, and photoelectric devices because of the large oscillator strength of the lowest-energy singlet transition, efficient luminescence, pronounced delocalization of the excited state, and balanced charge transport. The fabricated OFETs showed a photoelectric response for wavelengths shorter than 530 nm and simultaneously EL in the transistor channel, with a maximum at ~570 nm. The devices demonstrated an EL external quantum efficiency (EQE) of ~1.4% and a photoelectric responsivity of ~0.7 A W^–1^, which are among the best values reported for state-of-the-art organic light-emitting transistors and phototransistors, respectively. We anticipate that our results will stimulate the design of efficient materials for multifunctional organic optoelectronic devices and expand the potential applications of organic (opto)electronics.

## 1. Introduction

Organic electronics is a rapidly developing field of science and technology. Organic semiconductor materials, in particular conjugated oligomers, have been actively studied for recent decades as an alternative to silicon and other inorganic semiconductors with a high potential to simplify the production and reduce the cost of electronic devices, as well as to fabricate electronic devices with a wider range of functions. One of the basic organic electronic devices are organic field-effect transistors (OFETs), which are the basis for such promising devices as chemical sensors [1,2], organic light-emitting transistors (OLETs) [3,4] and organic phototransistors (OPTs) [5,6]. The latter two devices combine the transistor function with that of a light-emitting diode or photodiode, respectively, allowing for a decrease in weight, energy consumption, and production costs for displays and photosensing devices. Another new type of device proposed recently is a dual-function optoelectronic device combining light emission and light harvesting [7,8]. These devices are promising for indoor light harvesting, UV-absorbing smart windows, or multijunction solar cells [9,10]. However, these dual-function devices were implemented in the sandwich-like architecture, whereas the dual-function planar devices (i.e., working in the OFET geometry) have not been reported. Moreover, the studied dual-function optoelectronic devices were based on donor–acceptor blends so that the potential of single-material devices has not been explored.

Thiophene-phenylene co-oligomers (TPCOs) are among the most promising materials for efficient OLETs [11,12,13] and OPTs [14,15]. On the one hand, TPCOs can have fairly high and balanced mobility of charge carriers (electrons and holes), and, on the other hand, they are characterized by a high absorption coefficient and luminescence efficiency [16,17,18]. Bipolar charge transport (i.e., the electron and hole one) is favorable for OLETs and OPTs, making charge recombination/photogeneration more efficient [19,20]. Unfortunately, although prominent bipolar transport is predicted for many organic semiconductors, efficient injection of both carrier types is not frequently observed so that the devices typically show either only hole or electron transport [21]. The most probable reason is that the hole or electron injection in the device active layer is determined not only by the difference between the electrode work function and the highest occupied molecular orbital (HOMO) or lowest unoccupied molecular orbital (LUMO) energy, respectively; as commonly believed [21]; but also by an interface dipole at the electrode/active layer contact [22]. Such an interface dipole in TPCOs can be controlled by an appropriate choice of the terminal substituents of the donor or acceptor nature [23]. Recently, for a TPCO, 5,5′-bis [4-(trimethylsilyl)phenyl]-2,2′-bithiophene (TMS-PTTP-TMS), consisting of PTTP (P stands for phenyl and T stands for thiophene), and trimethylsylil (TMS) terminal substituents, bipolar charge transport was observed and assigned to the effect of TMS terminal substituents facilitating charge injection of electrons and holes in the device active layer [23]. Moreover, TMS-substituted TPCOs crystallize in platy crystals with strongly inclined molecules against their basal plane [16,24] because of the bulky TMS terminal substituents [25], and this inclination is beneficial for light outcoupling in single-crystal OLETs [24] and could enhance light absorption in single-crystal OPTs as well. However, the relatively short conjugation length of TMS-PTTP-TMS results in rather high LUMO/low HOMO energies [26], which can impede charge injection from the available electrodes. Moreover, TMS-PTTP-TMS showed a moderate photoluminescence quantum yield (PL QY) equal to 20 ± 2% both in solution and neat crystals [27]. On the other hand, a TMS-substituted TPCO with the longer conjugated chain, namely 5,5′″-bis [4-(trimethylsilyl)phenyl]-2,2′:5′,2″:5″,2′″-quaterthiophene, or TMS-P4TP-TMS (bearing two more thiophene rings in its center than TMS-PTTP-TMS, see Figure 1a), showed the doubled PL QY (44 ± 2% in solution) and the lower LUMO/higher HOMO energies [27], which are favorable for light-emissive devices and could facilitate charge injection, respectively. In addition, the longer conjugated length should facilitate lower exciton binding energy and easier separation of the photogenerated charges, which is advantageous for efficient OPTs. Summarizing, one can expect that TMS-P4TP-TMS can be promising as an active material for both OLETs and OPTs. Although TMS-P4TP-TMS was synthesized earlier [27], it was studied mainly as a dopant (for TMS-PTTP-TMS films) so that its crystal structure, charge-carrier mobility, electroluminescence, and performance in field-effect devices (OFET, OLET, and OPT) have not been reported.

Here, we show that an OFET can operate as a dual optoelectronic field-effect device, i.e., simultaneously combining light emission and photosensitivity. Our combined experimental and theoretical study highlighted that TMS-P4TP-TMS is promising for single-material light-emitting and photoelectric devices. Electroluminescence (EL) with an external quantum efficiency (EQE) of more than 1% was realized simultaneously with the photoelectric effect, corresponding to a responsivity of about 0.7 A W^–1^ at shorter wavelengths than EL. OFETs, combining the properties of light-emitting devices and phototransistors, open new possibilities for multifunctional organic optoelectronics.

## 2. Results and Discussion

### 2.1. Frontier Orbitals and Optical Properties from DFT Calculations

Figure 1 presents the optimized molecular geometry of TMS-P4TP-TMS along with HOMO and LUMO patterns for this molecule obtained from density functional theory (DFT) calculations. According to this figure, the HOMO and LUMO of the TMS-P4TP-TMS molecule are located at the P4TP core, whereas the TMS groups are practically not involved in the π-conjugated system. The HOMO and LUMO energies equal −4.76 and −1.96 eV, respectively. It is useful to compare these frontier orbital energies with those for its structurally close counterpart, TMS-PTTP-TMS, for which the HOMO level is deeper, whereas the LUMO is shallower (−4.9 eV and −1.7 eV, respectively, according to the earlier DFT calculations with the same functional and basis set [26]). These lower LUMO/higher HOMO energies of TMS-P4TP are assigned to the increased conjugation length as compared with TMS-PTTP-TMS (see below) and should facilitate efficient charge injection of both electrons and holes in TMS-P4TP-TMS. Our time-dependent DFT (TDDFT) calculations at the B3LYP/6-31G(d,p) level predict the lowest-energy absorption maximum at 493 nm (2.51 eV) and the PL maximum at 589 nm (2.11 eV). These wavelengths are larger than those for TMS-PTTP-TMS, i.e., both absorption and emission of TMS-P4TP-TMS are red-shifted as compared to TMS-PTTP-TMS, in line with the experimental data [27]. The oscillator strength for the S0-S1 transition (~2.2) in TMS-P4TP-TMS exceeds that for TMS-PTTP-TMS (~1.7) and explains both higher molar absorption (molar extinction 60,000 M^−1^ cm^−1^ at 432 nm) and PL QY in solution for the former [27]. With the use of the CAM-B3LYP functional and the same basis set, the lowest-energy absorption maximum is blue-shifted to 429 nm (2.91 eV), while the PL maximum is blue-shifted to 490 nm (2.53 eV) in good agreement with the experiment (see Section 2.5); meanwhile, the oscillator strength is virtually the same. The calculated conjugation length for TMS-P4TP-TMS (~13 Å) is also longer than that for TMS-PTTP-TMS (7.1 Å as calculated from the data from Ref. [26] concerning the corrections to the calculation method used therein) and TPCOs with the PTPTP core (~8.5–9 Å from Ref. [28]). The longer conjugation length is expected to result in higher excited state delocalization, which is favorable for the separation of photogenerated charges, specifically in OPT.

### 2.2. Crystal Growth and Analysis

To address the molecular packing for TMS-P4TP-TMS, its single crystals were vapor-grown. They did not have any specific habitus, especially those with lateral sizes greater than 600 μm (see Appendix A), though the crystals were platelike (the thickness was typically <5 μm). The color of the crystals varied from yellow to red with an increase in their thickness.

The x-ray diffraction data on TMS-P4TP-TMS single crystals indicated that the crystal class is monoclinic and the space group is C2/c. Table 1 presents the parameters of the unit cell, which includes four independent molecules (Appendix A). The molecular packing in TMS-P4TP-TMS crystals is similar to that in TMS-PTTP-TMS ones reported earlier [29]: TMS-P4TP-TMS molecules are organized in layers, forming a herringbone packing motif with face-to-face overlap [30,31] in the layer corresponding to the ab-plane, as shown in Figure 2a,b. The thickness of the layer is about 19.5 Å, and the herringbone angle is between 75.7° and 88.8°. The inclination angle of the molecules, i.e., the angle between the long molecular axis and the crystal basal plane (which coincides with the ab-plane, Appendix A), for all four independent molecules is ~53°. This pronounced inclination should be assigned to the effect of bulky TMS terminal substituents [24,29] and should enhance light outcoupling and absorption of light incident on the crystal face, which is beneficial for the higher efficiencies of electroluminescent and photosensing devices, respectively.

### 2.3. Charge-Carrier Mobility Calculations

The knowledge of the TMS-P4TP-TMS crystal structure allowed us to estimate computationally charge-carrier mobilities in crystal. According to the hopping model used herein (see Equation (1) and Ref. [21]), the charge-carrier mobilities are determined by the transfer integrals, the reorganization energies, and the distances between the molecular centers.

The transfer integrals for electrons and holes, *J_e_* and *J_h_*, respectively, for the crystal studied are presented in Figure 3. All considerable transfer integrals are observed within the molecular layer (slab) corresponding to the ab-plane. For holes, they are relatively small, the largest being comparable to the energy of thermal fluctuations of *k_B_T*~25 meV, where *k_B_* is Boltzmann constant and *T* is absolute temperature; while for electrons, the transfer integrals are much larger and amount up to 99 meV (>3 *k_B_T*). Different molecules within the unit cell are characterized by the slightly different transfer integrals with their neighbors. For both electrons and holes, charge transport is two-dimensional and is expected to be resilient to various defects [32], which are unavoidable in thin films. As also follows from Figure 3, large *J_e_* are observed for dimers with the face-to-face orientation of molecules (~80–100 meV), whereas those for dimers with the edge-to-face orientation are lower (below 25 meV), in line with our earlier results for other TPCOs [27,28]. On the contrary, relatively large *J_h_* are observed for dimers with edge-to-face orientation (~20–30 meV), whereas those for dimers with face-to-face orientation do not exceed 10 meV.

The reorganization energies for holes and electrons amount to 290 and 350 meV, respectively. These values are comparable with those reported for other TPCOs [28], and the larger reorganization energy for electrons than that for holes is also in line with the previous reports on TPCOs [28].

The charge-carrier mobilities calculated using the transfer integrals and the reorganization energies presented above equal 0.035 cm^2^V^−1^s^−1^ for holes and 0.19 cm^2^V^−1^s^−1^ for electrons. Despite the larger reorganization energy for electrons, the electron mobility is higher than the hole mobility because of much larger *J_e_* than *J_h_*. The charge-carrier mobilities are considerably lower as compared with high-mobility organic semiconductors based mainly on annulated conjugated moieties but typical for crystals of linearly conjugated oligomers [28].

### 2.4. Electrical Measurements

Figure 4 shows typical transfer characteristics for OFETs with thin-film (polycrystalline) and single-crystalline active layers. In both, the transfer characteristics are V-shaped, and the mobilities of electrons and holes are balanced. For holes, the experimental charge-carrier mobility is about twice lower than the calculated one, and for electrons, the experimental mobility is tenfold lower than the calculated one (see above). The lower experimental charge-carrier mobility values and the weaker difference between the hole and electron mobilities could be tentatively ascribed to the polycrystalline character of the thin film and charge-injection issues. We stress that the calculations were performed for a single crystal, and hence the calculated charge-carrier mobilities should be considered as a higher estimate for the experimental charge-carrier mobility. Although, as a rule, single-crystal OFETs exhibit high performance [33], for the TMS-P4TP-TMS single-crystal sample, the current values and hence the charge-carrier mobilities are much lower than for the thin-film one. This difference is assigned to the much thicker OFET active layer (i.e., the crystal thickness), which can hamper the injection of charges from the electrodes and their transport into the transistor channel through the active layer [34,35]. Table 2 shows maximal and average charge-carrier mobilities and average threshold voltages for thin-film and single-crystal OFETs.

The thin-film and single-crystal OFET active layers were studied using optical and atomic force microscopy; the corresponding data are given in Appendix A. As the thin films have a polycrystalline nature, they have a high roughness with an average value of ~22 nm, whereas the surface of the single crystals is molecularly smooth with an average roughness of ~0.05 nm. Optical images of two single-crystal OFET samples with different crystal thicknesses are presented in Appendix A. Figure 4b and Figure 5b present transport and EL data, respectively, for the OFET based on the thinner crystal (~1 μm), whereas the thicker crystal (>3 μm) did not work in the OFET (no current was observed in the channel at any voltage). One may expect that essentially thinner single crystals would show maximal performance; however, we could not grow such crystals.

### 2.5. Electroluminescence

Figure 5 summarizes our OLET data. Figure 5a presents thin-film OLET images obtained by superimposing a black-and-white image of the device under backlight and a colored image of the transistor channel while the gate voltage is changed stepwise. These images show thread-like EL regions in the transistor channel between the source and drain electrodes, partially extending out of the *W*x*L* rectangle. The latter clearly indicates that the transistor current partially flows beyond the *W*x*L* rectangle. Each thread (stripe) corresponds to a given gate voltage and hence to a certain position of the charge recombination zone at a moment of time during recording the transfer characteristic. Figure 5c shows EL spectra of thin-film and single-crystal OLETs, as well as PL and absorption spectra of TMS–P4TP–TMS in solution and thin film. The absorption spectrum in thin film is more structured, and its edge is red-shifted by 0.16 eV as a result of intermolecular interactions in TMS-P4TP-TMS. The EL spectrum of the single-crystal device is noisy because of the low EL intensity. The EL spectrum for the thin-film device and the PL spectrum in thin-film clearly show four bands at 529, 569, 614, and 670 nm (2.34, 2.18, 2.02, and 1.85 eV) with a solid-state red shift of ~0.12 eV relative to the PL spectrum in solution. The PL and EL spectra differ in the intensities of vibronic bands so that the blue part of EL is less intensive than that of PL. This difference could be explained mainly by EL self-absorption, as the PL was collected in the reflection geometry, whereas the EL passed the film thickness. Similarly, the essentially lower intensity of the EL 0-0 band (at 2.34 eV) in thin film as compared to that of PL in solution can also be assigned to EL self-absorption, which should somewhat reduce the EL EQE as well.

Note that compared with the TMS-PTTP-TMS devices [25], where the contribution of its self-dopant (e.g., TMS-P4TP-TMS) to the EL spectrum is significant, we did not observe the presence of self-dopants in both the EL and PL spectra of deeply purified TMS-P4TP-TMS. However, a pronounced red flank in the PL spectra was observed (not shown) in films prepared from raw TMS-P4TP-TMS (i.e., before high-vacuum sublimation). Therefore, in contrast to TMS-PTTP-TMS [27], TMS-P4TP-TMS can be purified from the self-dopants so that they have no impact on its luminescent properties.

Figure 5d shows EL EQE as a function of the gate voltage. The EL EQE reaches its maximum almost at the minimum of the transfer characteristic corresponding to the bipolar transistor mode. As clearly seen in Figure 5a, EL is observed in the channel, which is a solid evidence of bipolar transport in the OLET channel [36]. The EQE was estimated from the measured absolute EL power, and its maximal value reached 1.40 ± 0.15%. The EQE value of the TMS-P4TP-TMS OLET is among the highest ones reported for the state-of-the-art OLETs [37,38,39,40]. The corresponding luminous efficiency reached 4.3 ± 0.4 cd A^–1^, being inferior only to the record-performing OLET equipped with an optimized light outcoupling accessory [41].

EL was also observed for single-crystal samples, as shown in Figure 5b, but the efficiency was almost twice lower (EQE = 0.84 ± 0.09%). This lower EQE is attributed to the waveguiding effect so that the major part of EL comes to the crystal edges and exits at them and defects, specifically at the growth steps on the crystal surface, as was observed as oblique EL stripes in the inset (Figure 5b). Nevertheless, EL from the channel is also distinguished in this image as vertical stripe-like regions (perpendicular to the current direction). Note that among many other bipolar single-crystal OLETs [42], TMS-P4TP-TMS crystal is one of the most suitable for efficient light outcoupling as the waveguiding effect is weakened because of the high inclination of molecules to the crystal basal plane [28].

### 2.6. Photoelectric Effect

To study the photoelectric effect, the OFET samples were illuminated with modulated monochromatic light (Appendix A) so that the drain current, *I*_D_, changed as shown in Figure 6a. The photocurrent, *I*_ph_, was calculated as the difference in the *I*_D_ value with and without illumination. Figure 6b shows normalized photocurrent, *I*_ph_/*I*_D_ (where *I*_D_ is the drain current in the dark), as a function of the gate voltage for the thin-film OFET. The photocurrent reaches its maximum near *V*_G_ = −3 V, corresponding to the bipolar charge transport mode (see Figure 5d). Figure 6c illustrates the responsivity spectrum, which describes the efficiency of converting the incident photons into the photocurrent, and the EL spectrum of the same device. As follows from Figure 6c, the spectra almost do not overlap, i.e., the EL and the photoelectric response occur in the different spectral regions. Note that at wavelengths below 450 nm, *R* increases with the photon energy despite the non-monotonic behavior of the absorption spectrum (Appendix A), i.e., *R* is not proportional to the optical absorption. This increase in *R* can be explained by the contribution of the excess photon energy to charge separation. Indeed, the probability of exciton dissociation to separated charges is greater for excitons generated from photons with the energy exceeding the optical gap of the material (“hot excitons”) [43]. The larger this excess energy, the larger the exciton dissociation probability and the larger the photocurrent. Such a behavior was observed experimentally in Refs [44,45]. In summary, our thin-film OFETs show the maximal normalized photocurrent *I*_ph_/*I*_D_ = 1.1 and responsivity *R* = 0.71 A W^–1^ at 375 nm, which is comparable to the *R* values reported for other thin-film OPT under similar incident light conditions (wavelengths and intensity) [46,47,48].

Figure 6d shows that the normalized photocurrent (*I*_ph_/*I*_D_) increases logarithmically with the incident light optical power, *P*_opt_. Such a dependence is a signature of a photovoltaic effect [49]. In general, two different types of photoelectric effects can occur in the OFET active layer—photoconductivity and photovoltaic effects [5,50]. The photovoltaic effect is characterized by a logarithmic dependence on the incident optical power and is caused by a shift in the threshold voltage because of the large number of trapped photogenerated charges in the channel [51,52,53,54]. The traps can be assigned to the polycrystalline structure of the active layer.

The photoelectric effect was also observed for the single-crystal OFETs, but its efficiency was lower: the maximal responsivity at a wavelength of 375 nm was estimated as *R* = 0.04 A W^–1^, and the maximal normalized photocurrent was *I*_ph_/*I*_D_ = 0.4. The lower photoelectric performance can be attributed to the inner filter effect: because of the high thickness of the single crystal, a major part of incident photons are absorbed in the active layer and do not reach the transistor channel near the interface between the active layer and the gate dielectric. However, the shelf stability of the single-crystal devices was higher than that of the thin-film ones; specifically, the photocurrent remained almost unchanged after more than 100 days of storing the sample in a glovebox with an inert atmosphere (Appendix A).

## 3. Materials and Methods

### 3.1. Quantum Chemical Calculations

DFT calculations were performed in the GAMESS package [55,56] using the B3LYP functional and 6-31G(d,p) basis sets. Time-dependent DFT (TDDFT) calculations were conducted using B3LYP and CAM-B3LYP DFT functionals with the same basis set. Conjugation lengths were calculated as $l_{c}=\sqrt{3\sum_{i}c_{i}^{2}r_{i}^{2}}/\sqrt{\sum_{i}c_{i}^{2}}$, where *c* are the coefficients of the *i*-th atom wavefunctions at the HOMO in line with Ref. [28]. Hirshfeld surface analysis was performed using CrystalExplorer 17.5 software with the implemented TONTO package for DFT calculations [57] at the B3LYP/6-31G(d,p) level.

Charge-carrier mobility was calculated within the hopping model [21] using the Marcus formula [58] for the charge transfer rate:(1)$$k=\frac{2\pi }{\hbar }J^{2}{\left(\frac{1}{4\pi \lambda k_{B}T}\right)}^{1/2}exp\left(-\frac{{\left(∆E-\lambda \right)}^{2}}{4\lambda k_{B}T}\right)$$
where *ħ* is the reduced Planck constant and Δ*E* = *E*_2_ − *E*_1_ is the difference in the energies of the charge carrier between the initial and final sites. For one-component crystals, all molecules are similar (neglecting the possible difference in molecular environment), and hence the energy of the charge carrier on them is the same, i.e., Δ*E* = 0. The reorganization energy (*λ*) was approximated by its inner-sphere part, which is typically considered much larger than the outer-sphere part. The *λ* values were calculated according to the four-point scheme [21] from the energies of the molecule in four states: the neutral state in its optimized geometry (*E_N_*), the neutral state in the optimized geometry of the charged state (*E_N_^*^*), the charged state in its optimized geometry (*E_C_*), and the charged state in the geometry of the neutral state (*E_C_*^*^). The energy difference between the former two states, $\lambda_{1}=E_{N}^{*}-E_{N}$, describes the energy relaxation of the molecule that has lost a charge carrier, whereas the energy difference between the latter two states, $\lambda_{2}=E_{C}^{*}-E_{C}$, describes the energy relaxation of the molecule that has accepted a charge. The total reorganization energy is $\lambda =\lambda_{1}+\lambda_{2}=\left(E_{N}^{*}-E_{N}\right)+\left(E_{C}^{*}-E_{C}\right)$. The transfer integrals, *J*, were calculated using a home-written code based on the dimer projection method [59]. For calculations of *J* and *λ* values, DFT at the B3LYP/6-31G(d,p) level was utilized. Finally, charge-carrier mobilities were calculated from the rate constants (Equation (1)), averaged over non-identical molecules in the unit cell, using the Einstein-Smoluchowski relation (see, e.g., Ref. [60]).

### 3.2. Experimental Methods

The synthesis of TMS-P4TP-TMS was described earlier [27]. The synthesized product was sublimed multiple times with a high-vacuum purification apparatus [27].

The x-ray diffraction data for TMS-P4TP-TMS single crystal (CCDC 2050388) were collected at room temperature (295 K) with a diffractometer (STOE, Chicago, IL, USA), detector Pilatus (100 K) using the rotation method, a collimating mirror, and Cu Kα (1.54086 Å) radiation. The details for *x*-ray diffraction analysis are given in Appendix A.

Absorption spectra in thin films were measured using a spectrophotometer Lambda 25 (PerkinElmer, Shelton, CT, USA), and their PL spectra were recorded in the reflection geometry using a fiber spectrometer YSM-8104-08 (YIXIST, Hangzhou, China) with PL excitation at a wavelength of 440 nm. Thin films for optical spectroscopy were prepared on glass substrates by thermal vacuum evaporation, as described in the next section.

OFET samples were prepared almost in the same way as described in Refs. [23,24,28]. Two types of OFET devices were fabricated: thin-film and single-crystal ones.

#### 3.2.1. Fabrication of Thin-Film OFETs

OFET samples were prepared on silicon substrates covered by a 300 nm thick oxide layer and a 50 nm thick poly(methyl methacrylate) (PMMA) layer. PMMA layers were deposited onto the Si/SiO_2_ substrates by spin coating at 3000 rpm for two minutes from 45 µL of 10 g L^−1^ solution in toluene, and then it was subsequently annealed on a hot plate at 70 °C for 20 h and at 110 °C for 3 h. A 50 nm thick TMS-P4TP-TMS active layer was deposited on the top of the PMMA layer by thermal vacuum evaporation in a vacuum chamber (Univex 300G, Leybold, Cologne, Germany). Asymmetrical electrodes were used as drain and source contacts. As a hole-injecting electrode, a 50 nm thick Ag layer with a 10 nm thick interlayer of MoO_3_ was used, and as an electron-injecting electrode, an 80 nm thick Ca layer with a 4 nm thick interlayer of CsF was used. The materials were thermally evaporated in a vacuum chamber through shadow masks, which defined *L* = 25 µm and *W* = 1 mm.

#### 3.2.2. Fabrication of Single-Crystal OFETs

TMS-P4TP-TMS single crystals were grown using a physical vapor transport technique in laminar helium flow for five days (see details in Ref. [27]). The grown single crystals were transferred onto silicon substrates, covered by a 300 nm thick oxide layer and a 50 nm thick PMMA layer. After that, CsF/Ca and MoO_3_/Ag top electrodes were evaporated on the crystals using shadow masks exactly in the same way as it was done for the thin-film OFET samples. The channel length and width were *L* = 25 µm and *W* = 100 µm for the single-crystal devices.

#### 3.2.3. OFET Characterization

Electrical measurements were performed in a nitrogen atmosphere with a source meter (Keithley 2636A, Keithley Instruments, Solon, OH, USA) using a probe station (ProbeStation 100, Printeltech, Moscow, Russia). The charge-carrier mobility and threshold voltages were evaluated according to Shockley’s equations in the linear and saturation regimes: $I_{D}=\frac{W}{L}\mu_{lin}C\left(V_{G}-V_{T}\right)V_{D}{;\;I}_{D}=\frac{W}{2L}\mu_{sat}C{\left(V_{G}-V_{T}\right)}^{2}$, where *μ*_lin_ and *μ*_sat_ are the charge-carrier mobilities in the linear and saturation regime, respectively; *C* = 11.5 nF cm^−2^ is the gate insulator capacitance per unit area for a SiO_2_ layer with a thickness of 300 nm and a dielectric constant of 3.9; *V*_T_ is the threshold voltage.

#### 3.2.4. Electroluminescence

EL was characterized as described in Refs. [24,25]. The experimental schematic for EL and photoelectric effect studies is presented in Appendix A. Briefly, light emission images were captured using a microscope equipped with a 10× objective with numerical aperture of 0.28 and focal distance of 20 mm (M Plan Apo 10×, Mitutoyo, Kawasaki, Japan) and a CCD-camera (Infinity 3, Lumenera, Ottawa, ON, Canada) at exposition times of 20–60 s during the transfer characteristics measurements. The relative EL intensity was calculated as a sum of the intensities of pixels belonging to the channel area in a captured light-emission image. The EL spectra were measured using a custom-made spectrograph. The absolute EL power was measured by a photodetector based on a wide-area silicon photodiode with the tabulated responsivity spectrum (S1227-66BR, Hamamatsu Photonics, Hamamatsu, Japan) for estimation of the OFET luminance and EL EQE.

#### 3.2.5. Photoelectric Effect

To characterize the photoelectric effect in OFETs, the photocurrent measurements were performed under incident illumination. As the illumination source, a xenon lamp with a monochromator was used. Using an optical system, the illumination was applied to the OFET channel from above (as shown in Appendix A). The intensity of the incident optical radiation depends on the wavelength and was in the range of 0.5–5.0 W m^–2^. Using a mechanical chopper, the incident illumination was periodically turned on and off with a period of 2.0 s. Under these conditions, the drain current (*I*_D_) in the OFET samples was measured so that rectangular-shaped drain current transients were recorded (Figure 6a), from which the values of the drain current under illumination and in the dark were distinguished, and the photocurrent (*I*_ph_) was calculated as their difference. The photocurrent as a function of the drain and gate voltages was measured with a voltage sweep rate of 3 V s^–1^ in the range from 30 to –60 V at a bias of –50 V under monochromatic incident illumination with an intensity of 2.8 W m^2^ and at a wavelength of 420 nm. The photocurrent spectra were recorded at constant drain and gate voltages *V*_D_ = *V*_G_ = –50 V, while the wavelength was tuned from 380 to 800 nm with a 5 nm step and about 1.3 nm s^–1^ average speed. The responsivity *R* values were calculated as *R* = *I*_ph_/*P*_opt_, where *P*_opt_ is the incident optical power.

#### 3.2.6. Optical and Atomic Force Microscopy

The OFET active layer was characterized using optical and atomic force microscopy. The TMS-P4TP-TMS single crystals were characterized with the use of AxioImager A2m optical microscope with 10×, 20× and 100× objectives (Carl Zeiss AG, Oberkochen, Germany) working in circular polarized light–differential interference contrast (C-DIC) regime to determine the lateral dimensions and estimate crystallinity and surface homogeneity. An atomic force microscope NTEGRA Spectra (NT-MDT, Moscow, Russia) with cantilevers of HQ:XSC11 series (MikroMasch, Sofia, Bulgaria) was used in the semi-contact mode to determine the thickness of the thin films and single crystals and the morphology of their surfaces.

## 4. Conclusions

In this work, charge transport, electroluminescence, and photoelectric properties of a thiophene-phenylene co-oligomer, TMS-P4TP-TMS, were studied in field-effect devices. An organic ambipolar field-effect transistor showing both bright electroluminescence and a pronounced photoelectric effect was demonstrated. The devices showed electroluminescence peaked at 570 nm with an external quantum efficiency of 1.4%, which is one of the best values for organic light-emitting transistors. The photoelectric response was observed at wavelengths corresponding to the optical absorption of the co-oligomer with a responsivity of about 0.7 A W^–1^, so that electroluminescence and photocurrent spectra almost do not overlap. The high electroluminescence and photoelectric performance are explained mainly by the balanced electron and hole mobilities in combination with the pronounced optical absorption and emissive properties of TMS-P4TP-TMS. We anticipate that combination of the functionalities of light-emitting devices and phototransistors can be used in smart devices, e.g., (micro)displays with the brightness adjusting to the ambient light conditions. Such multifunctional organic field-effect transistors open up new applications for organic optoelectronics.

## Figures and Tables

**Figure 1 molecules-29-02533-f001:**
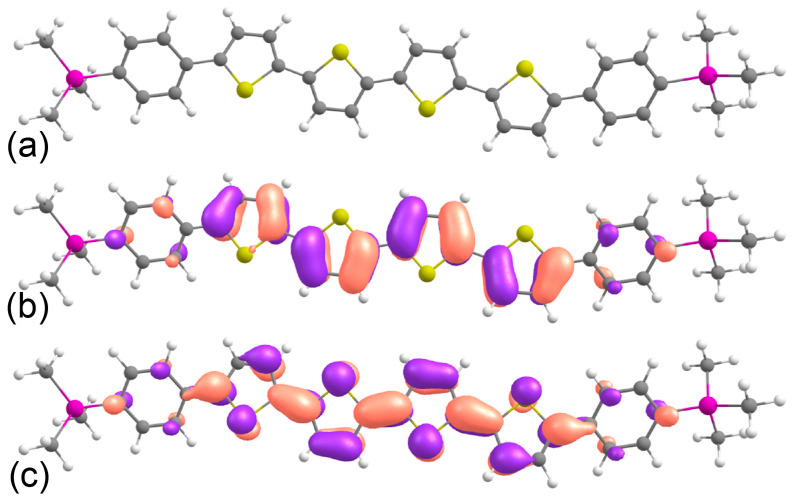
Calculated equilibrium geometry (**a**), HOMO (**b**) and LUMO (**c**) patterns of TMS-P4TP-TMS molecule. Hydrogen atoms are shown in white, carbon in grey, sulfur in yellow, and silicon in purple.

**Figure 2 molecules-29-02533-f002:**
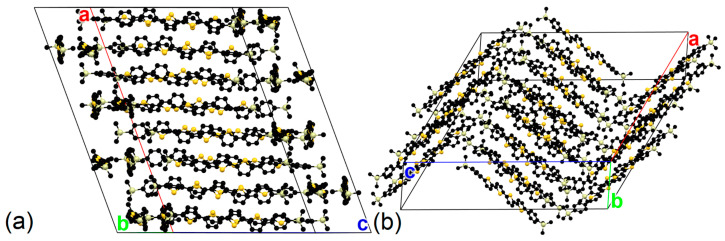
Crystal structure of TMS-P4TP-TMS in two projections: view on the herringbone motif (**a**) and on the layered packing (**b**). Black spheres denote carbon atoms, yellow spheres denote sulfur atoms, and white spheres denote silicon atoms; hydrogen atoms are not shown, ***a***, ***b***, and ***c*** are the lattice vectors.

**Figure 3 molecules-29-02533-f003:**
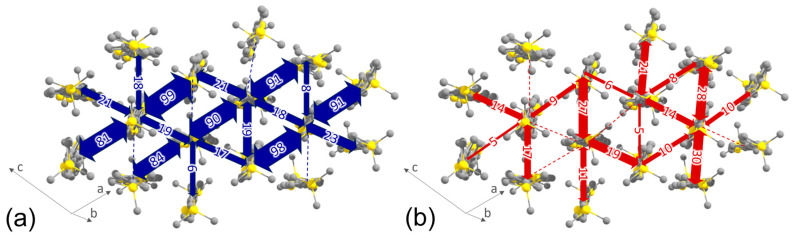
Charge transport directions for electrons (**a**) and holes (**b**) in the TMS-P4TP-TMS crystal; all the presented molecules belong to the same molecular layer. The thickness of the arrows connecting various molecules depicts the magnitude of the corresponding transfer integrals; the latter are also labeled (in meV). Transfer integrals below 5 meV are shown with dashed lines and are not labeled. Arrows in the left bottom corners of the panels denote the lattice vectors ***a***, ***b***, and ***c***.

**Figure 4 molecules-29-02533-f004:**
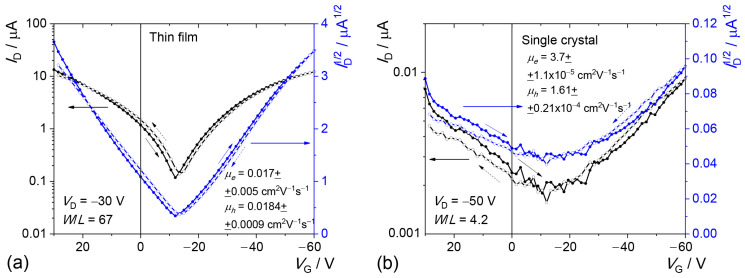
Transfer characteristics for thin-film (**a**) and single-crystal (**b**) OFET. Blue lines denote the transfer characteristics in terms of square root of the drain current.

**Figure 5 molecules-29-02533-f005:**
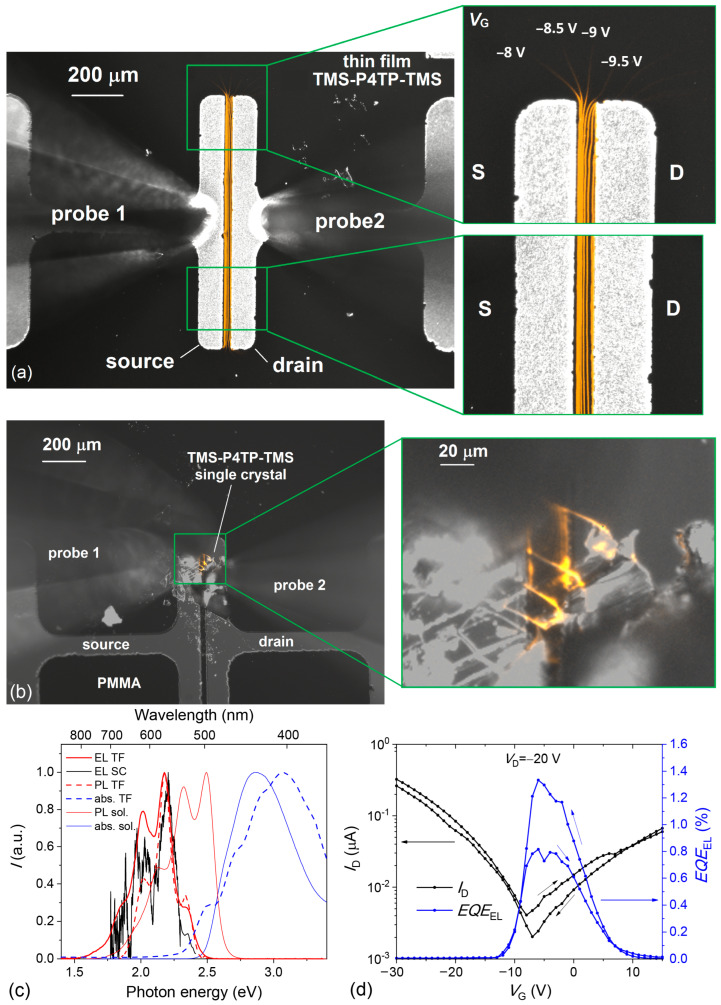
(**a**) Image of thin-film OLET, EL image (orange) was captured at an exposure of 18 s, while gate voltage *V*_G_ was changed from –12 to –8 V with step 0.5 V (9 points) at drain voltage *V*_D_ = –30 V. The EL image is superimposed with the OLET image captured under backlight. Letters S and D denote source and drain electrodes, correspondingly. (**b**) Image of a single-crystal OLET with its EL image under operation. (**c**) EL spectra of thin-film (TF) and single-crystal (SC) OLETs, PL, and absorption spectra of TMS-P4TP-TMS in thin film and in solution (spectra in solution are represented in Ref. [27]). (**d**) Drain current *I*_D_ and EL EQE versus *V*_G_ for the thin-film OLET. Original images of OLETs under backlight and in dark are given in Appendix A.

**Figure 6 molecules-29-02533-f006:**
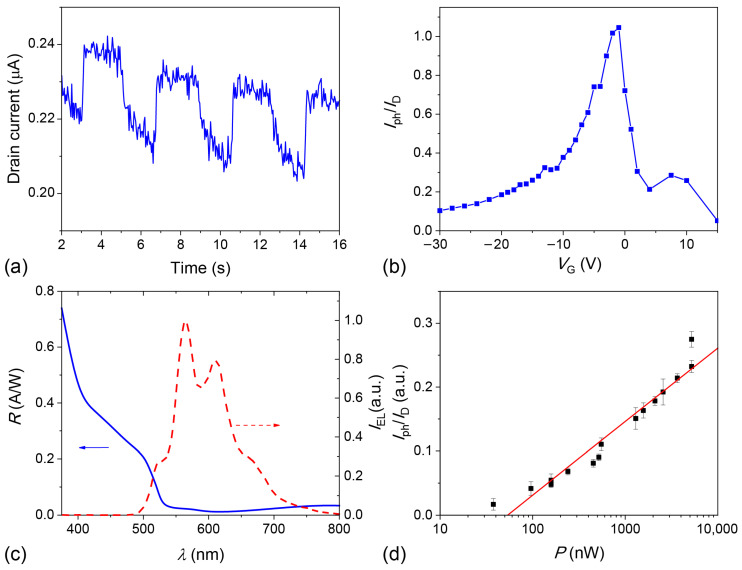
Photoelectric effect in the thin-film devices: (**a**) typical drain current, *I*_D_, vs time at constant drain and gate voltages under chopped incident illumination, (**b**) normalized photocurrent, *I*_ph_/*I*_D_, vs the gate voltage at *V*_D_ = –20 V, (**c**) responsivity spectrum at *V*_G_ = –30 V and *V*_D_ = –40 V (blue solid line); the EL spectrum (red dashed line) taken from Figure 5c is shown for reference, (**d**) normalized photocurrent as a function of the incident optical power at *λ* = 530 nm, *V*_D_ = –20 V and *V*_G_ = –10 V; the red line is a fit to the experimental data with a logarithmic function.

**Table 1 molecules-29-02533-t001:** Unit cell parameters of TMS-P4TP-TMS single crystals.

*a*, Å	*b*, Å	*c*, Å	*α*, deg	*β*, deg	*γ*, deg	Z	Space Group
43.4110(10)	14.8981(5)	45.184(2)	90	115.795(3)	90	32	C 2/c

**Table 2 molecules-29-02533-t002:** Maximal and average charge-carrier mobilities and average threshold voltages for OFETs with CsF/Ca and MoO_3_/Ag top electrodes. The maximal and average values were calculated among 20 devices for thin-film OFETs and among three devices for single-crystal ones.

Active Layer	Charge Carriers	Max. Mobility, cm^2^ V^–1^ s^–1^	Aver. Mobility, cm^2^ V^–1^ s^–1^	Aver. Threshold Voltage, V
Thin film	electrons	0.0237	0.0092 ± 0.0010	19 ± 6
	holes	0.0193	0.0108 ± 0.0009	–10 ± 4
Single crystal	electrons	1.0·10^–4^	(5.0 ± 1.7)·10^–5^	–0.3 ± 16
	holes	1.61·10^–4^	(1.06 ± 0.22)·10^–4^	5 ± 10

## Data Availability

The data presented in this study are available in Appendix A.

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
