# Peer review of "Dual Optoelectronic Organic Field-Effect Device: Combination of Electroluminescence and Photosensitivity"

_molecules, 2024, doi:10.3390/molecules29112533_

Round 1

Reviewer 1 Report

Comments and Suggestions for Authors

The authors present a study related to Dual optoelectronic organic field-effect device: combination of electroluminescence and photosensitivity. They show results of your device's optical properties. I consider that the manuscript can be considered in its present form.

I suggest the authors only adjust the document to the journal format and improve the conclusions section.

Author Response

Thank you very much for taking the time to review this manuscript. Please find response below and the corresponding revisions/corrections highlighted in the re-submitted files

Comments 1:

The authors present a study related to Dual optoelectronic organic field-effect device: combination of electroluminescence and photosensitivity. They show results of your device's optical properties. I consider that the manuscript can be considered in its present form.

I suggest the authors only adjust the document to the journal format and improve the conclusions section.

Response 1:

In the revised manuscript, we adjusted the manuscript to the journal format and revised conclusions to make it clearer and more grounded.

Reviewer 2 Report

Comments and Suggestions for Authors

Trukhanov et al. studied both computationally and experimentally thin-films and single-crystals organic field-effect transistors (OFET) based on an extended thiophene-phenylene co-oligomer (TPCO) showing suitable electronic properties and balanced charge transport allowing to combine light-emitting capability (electroluminescence, EL) and photoelectric response in order to produce multifunctional organic optoelectronic devices. Overall, the methodologies and the experimental results are sufficiently well detailed. Noticeably, the measured photelectric responses and EL EQE values obtained for the thin-film OFET of the TMS-P4TP-TMS oligomer were found to approach the performance of other state-of-the-art organic light-emitting transistors and phototransistors. For these reasons, in my opinion the presented work would deserve publication in Molecules. However, some concerns relate to those conclusions (i.e. high delocalization of excited states, large oscillator strength, efficient luminescence) which were drawn on missing data. The impact of the work would thus considerably improve by some changes that I would recommend to address prior to submission:

1)      Please, add the results of the DFT and TD-DFT calculations. The author detailed the charge localization and delocalization along the molecular structure in the dedicated section (paragraph 2.1), also comparing these results to the experimental absorption and emission wavelengths and to the analogous TMS-PTTP-TMS, reported in previous investigations (refs. 21-22), but they did not show the relative frontier molecular orbitals either in the Manuscript or in the Supporting Information. In absence of any chart showing the investigated structure, inserting a new figure depicting the orbitals mainly involved in the allowed S0àS1 transition might be beneficial for the data presentation and discussion.

2)      The predicted lowest-energy absorption and emission maxima reported on page 3 of the Manuscript resulted to be shifted of approximately 60 and 80 nm, respectively, from the experimental energies shown only for TMS-P4TP-TMS in THF solution (Fig. S5). Similarly, the simulation on the charge-carrier mobility based on transfer integrals and reorganization energies provided very different results (i.e. tenfold higher charge-carrier mobility as for the electrons) from the experimental values reported in Table 2. I wonder whether the chosen method, namely the B3LYP/6-31G(d,p), could be improved/changed to better reproduce the experimental findings.

3)      In several parts of the Manuscript the authors stressed the efficient luminescence of TMS-P4TP-TMS, but the fluorescence quantum yield value (either in solution, thin-film or single crystal) was never reported. I would suggest to explicit to the reader at least one of these PL quantum yields.

4)      In Fig. 4c the electroluminescence spectra of the thin-film and single-crystal OLETs (solid-state materials) were compared to the fluorescence spectrum recorded for TMS-P4TP-TMS in THF solution.  I do not question the inner-filter effect phenomenon which was invoked to explain the different intensity of the peaks. However, I would more straightforwardly compare the electroluminescence profile to the emission/absorption spectra measured for a reference TMS-P4TP-TMS thin film and/or single crystal considering the profound differences of the materials in terms of electronic properties and intermolecular interactions when switching from solution to solid-state.  

Minor issues:

 -          Please add some details about the microscopy image acquisition method in the caption of Fig. S1

 -     On page 4 and in the caption of Fig. S4 (page 5, Supporting Information, Hirshfeld surface analysis paragraph) the compound label is misspelled: please correct TMS-PTTTTP-TMS with TMS-P4TP-TMS. The cited Fig.S3 should be S4 in the same paragraph. Also, Figure S4 is never mentioned along the text of the Manuscript.

-         I would rephrase “lower/higher LUMO/HOMO energies” with “lower LUMO / high HOMO energies” on pages 1 and 3 of the Manuscript.

-        Similarly, on page 5, lines 4-5 of 2.4 Electrical measurements paragraph, please clarify the sentence as follows: “The experimental charge-carrier mobility is about twice lower for holes and tenfold lower for electrons if compared to the calculated ones (see above).”

-        There is no Fig. S6c in the Supporting Information despite the fact that it is cited on page 14 of the Manuscript (3.2.5 Photoelectric effect). Please, insert the right number.

-    Figure 5d on page 11 of the Manuscript. Some of the experimental normalized photocurrent values obtained at lower incident power (P < 1000 nW) appear to have been purposely excluded from the logarithmic fitting, red line. Please, justify this choice. 

Comments on the Quality of English Language

English language needs to be revised.

Author Response

Thank you very much for taking the time to review this manuscript. Please find response below and the corresponding revisions/corrections highlighted in the re-submitted files. In this response, criticisms are given in italics and followed by our detailed point-to-point responses (the positive remarks are omitted for brevity). Apart from the changes to the manuscript made to address these issues, its minor revision was done to improve grammar, style, wording, to fix typos, etc. In addition, we corrected wavelengths of EL spectra by shifting them by 5 nm after the additional check of wavelength calibration of our spectrometer used for EL measurements. The page and line numbers are given according to the revised version of the manuscript (version with no highlighted changes).

Comments 1:

However, some concerns relate to those conclusions (i.e. high delocalization of excited states, large oscillator strength, efficient luminescence) which were drawn on missing data.

Response 1:

In the revised manuscript, we revised conclusions by replacing the fifth sentence in Section 4 (page 13, lines 448-450):

“The high electroluminescence and photoelectric performance are explained by the balanced electron and hole mobilities, large oscillator strength of the lowest singlet optical transition, efficient luminescence, and high delocalization of the excited state because of a relatively long conjugated length of the oligomer.”

by:

“The high electroluminescence and photoelectric performance are explained mainly by the balanced electron and hole mobilities in combination with high optical absorption and emissive properties of TMS-P4TP-TMS.”

To substantiate high optical absorption of TMS-P4TP-TMS, we added the numerical value of molar extinction in Section 2.1 (page 3, line 115). To additionally support the promising optical properties of TMS-P4TP-TMS, we added the following sentence in Section 2.2 (page 4, lines 145-148):

“This pronounced inclination should be assigned to the effect of bulky TMS terminal substituents [24,29] and promotes light outcoupling and absorption of light incident on the crystal face, which should facilitate the higher efficiencies of electroluminescent and photosensing devices, respectively.”

Comments 2:

The impact of the work would thus considerably improve by some changes that I would recommend to address prior to submission:

  • Please, add the results of the DFT and TD-DFT calculations. The author detailed the charge localization and delocalization along the molecular structure in the dedicated section (paragraph 2.1), also comparing these results to the experimental absorption and emission wavelengths and to the analogous TMS-PTTP-TMS, reported in previous investigations (refs. 21-22), but they did not show the relative frontier molecular orbitals either in the Manuscript or in the Supporting Information. In absence of any chart showing the investigated structure, inserting a new figure depicting the orbitals mainly involved in the allowed S0àS1 transition might be beneficial for the data presentation and discussion.

Response 2:

In the revised manuscript, we added Fig. 1b,c,d in Section 2.1 to show the optimized molecular geometry of TMS-P4TP-TMS along with its HOMO and LUMO patterns.

Comments 3:

2)      The predicted lowest-energy absorption and emission maxima reported on page 3 of the Manuscript resulted to be shifted of approximately 60 and 80 nm, respectively, from the experimental energies shown only for TMS-P4TP-TMS in THF solution (Fig. S5). Similarly, the simulation on the charge-carrier mobility based on transfer integrals and reorganization energies provided very different results (i.e. tenfold higher charge-carrier mobility as for the electrons) from the experimental values reported in Table 2. I wonder whether the chosen method, namely the B3LYP/6-31G(d,p), could be improved/changed to better reproduce the experimental findings.

Response 3:

We agree that the discrepancy between calculated and experimental energies of absorption and photoluminescence maxima can be resolved via appropriate choice of DFT functional and basis set. In the course of this revision, we performed calculations using CAM-B3LYP and the same basis set. As a result, the lowest-energy absorption maximum and emission maximum were found to be at 429 and 490 nm, respectively, which are very close to the experiment. In the revised manuscript, we added the following on page 3, lines 116-120:

“With the use of the CAM-B3LYP functional and the same basis set, the lowest-energy absorption maximum is blueshifted to 429 nm (2.91 eV), while the PL maximum is blueshifted to 490 nm (2.53 eV) nm in good agreement with the experiment (see Fig. 5c); meanwhile, the oscillator strength is virtually the same.”

The difference of the simulation and experimental charge-carrier mobility values was explained in the initial manuscript by charge injection issues of at the source/drain contacts and polycrystalline character of the film: «The lower experimental charge mobility values and the weaker difference between the hole and electron mobilities could be tentatively ascribed to the polycrystalline character of the thin film and charge-injection issues.» (the fourth sentence in Section 2.4) We stress that the calculations were performed for single crystal, whereas the experiments were done for polycrystalline films, and hence the calculated charge-carrier mobilities should be considered as a higher estimate for the latter. For this reason, we consider that the calculations of the transfer integrals presented in the initial manuscript are relevant. In the revised manuscript, we added following sentence in Section 2.4 (page 5, lines 198-200):

“We stress that the calculations were performed for single crystal, and hence the calculated charge-carrier mobilities should be considered as a higher estimate for the experimental charge mobility.”

Comments 4:

3)      In several parts of the Manuscript the authors stressed the efficient luminescence of TMS-P4TP-TMS, but the fluorescence quantum yield value (either in solution, thin-film or single crystal) was never reported. I would suggest to explicit to the reader at least one of these PL quantum yields.

Response 4:

In the revised manuscript (the 2nd paragraph, Introduction), we added the values of photoluminescence quantum yield in solution for TMS-PTTP-TMS (page 2, line 75) and TMS-P4TP-TMS (page 2, line 78), from our earlier work: “Moreover, TMS-PTTP-TMS showed a moderate photoluminescence quantum yield (PL QY) equal to 20 ± 2% both in solution and neat crystals. On the other hand, a TMS-substituted TPCO with the longer conjugated chain, namely 5,5′″-bis[4-(trimethylsilyl)phenyl]-2,2′:5′,2″:5″,2′″-quaterthiophene, or TMS-P4TP-TMS (bearing two more thiophene rings in its center than TMS-PTTP-TMS, see Figure 1a), showed the doubled PL QY (44 ± 2% in solution) and the lower LUMO / higher HOMO energies,[27] which are favorable for light-emissive devices and could facilitate charge injection, respectively.”

Comments 5:

4)      In Fig. 4c the electroluminescence spectra of the thin-film and single-crystal OLETs (solid-state materials) were compared to the fluorescence spectrum recorded for TMS-P4TP-TMS in THF solution.  I do not question the inner-filter effect phenomenon which was invoked to explain the different intensity of the peaks. However, I would more straightforwardly compare the electroluminescence profile to the emission/absorption spectra measured for a reference TMS-P4TP-TMS thin film and/or single crystal considering the profound differences of the materials in terms of electronic properties and intermolecular interactions when switching from solution to solid-state. 

Response 5:

In the revised manuscript, we added absorption and photoluminescence spectra in thin film of TMS-P4TP-TMS in Figure 4c and Figure S5, and also a discussion to Section 2.5 (pages 6-7, lines 241-244): “The PL and EL spectra differ in the intensities of vibronic bands so that the red part of EL is less intensive than that of PL. We explain this difference by the EL self-absorption as PL was collected in the reflection geometry, whereas EL passed the film thickness.” 

In addition, we added the following sentence on comparison of the absorption spectra (page 6, lines 235-237):

“The absorption spectrum in thin film is more structured and its edge is redshifted by 0.16 eV as a result of intermolecular interactions of TMS-P4TP-TMS.”

The corresponding experimental details of thin-film optical characterization were added in the revised manuscript (section 3.2).

Comments 6:

Minor issues:

 -          Please add some details about the microscopy image acquisition method in the caption of Fig. S1

Response 6:

We added the following in the caption to Fig. S1: “The images were captured with the use of differential interference-contrast microscopy in circularly polarized light (AxioImager A2m, Zeiss with 10x objective).”

Comments 7:

 -     On page 4 and in the caption of Fig. S4 (page 5, Supporting Information, Hirshfeld surface analysis paragraph) the compound label is misspelled: please correct TMS-PTTTTP-TMS with TMS-P4TP-TMS. The cited Fig.S3 should be S4 in the same paragraph. Also, Figure S4 is never mentioned along the text of the Manuscript.

Response 7:

We replaced TMS-PTTTTP-TMS with TMS-P4TP-TMS and Fig. S3 with Fig. S4 in the Hirshfeld surface analysis paragraph and in the caption to Fig. S4.

Comments 8:

-         I would rephrase “lower/higher LUMO/HOMO energies” with “lower LUMO / high HOMO energies” on pages 1 and 3 of the Manuscript.

Response 8:

This is done in the revised manuscript.

Comments 9:

-        Similarly, on page 5, lines 4-5 of 2.4 Electrical measurements paragraph, please clarify the sentence as follows: “The experimental charge-carrier mobility is about twice lower for holes and tenfold lower for electrons if compared to the calculated ones (see above).”

Response 9:

In the revised manuscript, we rephrased this sentence in the following way: “For holes, the experimental charge-carrier mobility is about twice lower than the calculated one, and, for electrons, the experimental mobility is tenfold lower than the calculated one (see above).”

Comments 10:

-        There is no Fig. S6c in the Supporting Information despite the fact that it is cited on page 14 of the Manuscript (3.2.5 Photoelectric effect). Please, insert the right number.

The number of the cited figure in Section 3.2.5 (Figure 5a instead of Figure S6c) was corrected.

Comments 11:

-    Figure 5d on page 11 of the Manuscript. Some of the experimental normalized photocurrent values obtained at lower incident power (P < 1000 nW) appear to have been purposely excluded from the logarithmic fitting, red line. Please, justify this choice. 

Response 11:

The red line in Figure 5d was not correct in the initial manuscript by mistake, it was not the fit to the experimental data. In the revised manuscript, we replaced the incorrected red line in Figure 5d with a correct logarithmic fit to all the experimental normalized photocurrent values.

Reviewer 3 Report

Comments and Suggestions for Authors

At first glance, the work ( „Dual optoelectronic organic field-effect device: combination of electroluminescence and photosensitivity”) is quite good. However, upon closer analysis, serious concerns arise- primarily due to the need for novelty elements (the work essentially duplicates ideas described in [22]. The analytical part and the lack of sufficient experimental confirmation also raise great doubts.

  1. The introduction is written poorly, and it partially repeats the information described in the literature introduction of the work [22]. Moreover, outdated references dominate throughout the work. When it comes to works from the last five years, there are only 13 of them, but still, 9 of them are self-citations! This makes me doubt whether the described research will interest Materials readers.
  2. There is no description of the synthesis methods and no data confirming the purity of the test compound. The authors refer again to work [22] (the compound is the same as the one tested previously). But in the previous work one can read that :

„However, from our numerous unsuccessful attempts to purify the host material from the dopant above 99.99% (by using multiple vacuum sublimations, vapor crystal growth, solution recrystallization), we concluded that the dopant molecules are firmly embedded in the host crystal and do not form their own phase, which could be detected by X-ray diffraction or DSC”.  

So, in that work, the same Authors claimed that the byproduct content is beneficial (self-doping). Then why do they not refer to this fact in the current work? How do these byproducts impact the properties described in this work? Has this been taken into account in the DFT calculations? Etc.

3.       Paragraphs 2.2, 2.3, and 2.4 are generally a repetition (or continuation) of the research described in the previous work. Notably, the tested compound has already been used to construct an OFET device. This  means that the only novelty in this work is the research described in paragraphs 2.5 and 2.6.

 To summarize, I can not recommend publishing this manuscript in the Materials in its present form.

Author Response

Thank you very much for taking the time to review this manuscript. Please find response below and the corresponding revisions/corrections highlighted in the re-submitted files. Criticisms are given in italics and followed by our detailed point-to-point responses. Apart from the changes to the manuscript made to address these issues, its minor revision was done to improve grammar, style, wording, to fix typos, etc. The figure and reference numbers (where applicable) are given according to the original version of the manuscript, and the page and line numbers are given according to the revised version of the manuscript (version with no highlighted changes).

Comments 1:

At first glance, the work ( „Dual optoelectronic organic field-effect device: combination of electroluminescence and photosensitivity”) is quite good. However, upon closer analysis, serious concerns arise- primarily due to the need for novelty elements (the work essentially duplicates ideas described in [22]. The analytical part and the lack of sufficient experimental confirmation also raise great doubts.

Response 1:

We respectfully disagree with Reviewer concerning the novelty of our manuscript. Our arguments are given below.

Comments 2:

  1. The introduction is written poorly, and it partially repeats the information described in the literature introduction of the work [22]. Moreover, outdated references dominate throughout the work. When it comes to works from the last five years, there are only 13 of them, but still, 9 of them are self-citations! This makes me doubt whether the described research will interest Materials readers.

Response 2:

In the revised manuscript (Introduction), we added a brief review on dual organic optoelectronic devices combining functionality of photodetectors and LEDs, and reinforced it by several up-to-date references. In addition, we deleted not the most important self-citations (five, numbers 28-30, 33, 61 according to the original manuscript) and added more up-to-date references to the articles by other authors relevant to our manuscript.

Comments 3:

  1. There is no description of the synthesis methods and no data confirming the purity of the test compound. The authors refer again to work [22] (the compound is the same as the one tested previously). But in the previous work one can read that :

„However, from our numerous unsuccessful attempts to purify the host material from the dopant above 99.99% (by using multiple vacuum sublimations, vapor crystal growth, solution recrystallization), we concluded that the dopant molecules are firmly embedded in the host crystal and do not form their own phase, which could be detected by X-ray diffraction or DSC”.  

So, in that work, the same Authors claimed that the byproduct content is beneficial (self-doping). Then why do they not refer to this fact in the current work? How do these byproducts impact the properties described in this work? Has this been taken into account in the DFT calculations? Etc.

Response 3:

The detailed description of synthesis of TMS-P4TP-TMS is given in our earlier work [22]. Regarding the purity of TMS-P4TP-TMS, we did observe the presence of self-dopants (i.e., longer oligomers than TMS-P4TP-TMS) in the photoluminescence spectra of the raw material. However, self-doped TMS-P4TP-TMS in the aggreged state, e.g., in crystals (the data are not presented in the manuscript), did not show higher photoluminescence as in the case of TMS-PTTP-TMS self-doped by TMS-P4TP-TMS. Therefore, self-doping of TMS-P4TP-TMS seems to be not so beneficial for its luminescent properties in contrast to self-doped TMS-PTTP-TMS. Moreover, we did not study the chemical composition of the self-dopant of TMS-P4TP-TMS (this is out of the scope of the present study).  However, after multiple purification cycles by high-vacuum sublimation of TMS-P4TP-TMS the material purity became high enough so that the self-dopants were not identified in the luminescence spectra, specifically, as confirmed by the absence of a red flank in the EL spectra. In the revised manuscript, we added a relevant discussion in Section 2.5 (page 7, lines 247-253):

“Note that compared with the TMS-PTTP-TMS devices,[25] where the contribution of its self-dopant (e.g., TMS-P4TP-TMS) to the EL spectrum is significant, we did not observe the presence of self-dopants in both EL and PL spectra of deeply purified TMS-P4TP-TMS. However, a pronounced red flank in the PL spectra was observed (not shown) in films prepared from raw TMS-P4TP-TMS (i.e., before high-vacuum sublimation). Therefore, in contrast to TMS-PTTP-TMS,[27] TMS-P4TP-TMS can be purified from the self-dopants so that they do no impact essentially its luminescent properties.“

Concerning the last question of Reviewer, we did not take into account byproducts in our DFT calculations because of we succeeded to get rid of them and we do not know their chemical structure (as mentioned above).

Comments 4:

  1. Paragraphs 2.2, 2.3, and 2.4 are generally a repetition (or continuation) of the research described in the previous work. Notably, the tested compound has already been used to construct an OFET device. This means that the only novelty in this work is the research described in paragraphs 2.5 and 2.6.

Response 4:

We respectfully disagree with Reviewer  and believe that they misunderstood the novelty of the manuscript. Concerning the oligomer studied (TMS-P4TP-TMS), its crystal structure data (paragraph 2.2), charge-carrier mobility and measurements in OFET devices (paragraphs 2.3, 2.4) were not reported earlier. Although TMS-P4TP-TMS was synthesized in our earlier work, it has never been studied in OFETs as a single material, but only as a dopant for TMS-PTTP-TMS. To avoid such misunderstandings among the readers and emphasize the novelty of the data presented in the manuscript, we added the following sentence into Introduction (page 2, lines 84-87):

“Although TMS-P4TP-TMS was synthesized earlier [27], it was studied mainly as dopant (for TMS-PTTP-TMS films) so that its crystal structure, charge-carrier mobility, electroluminescence, and performance in field-effect devices (OFET, OLET and OPT) have not been reported.”

Moreover, in the revised manuscript, we added Figure 1a with the chemical structure of TMS-P4TP-TMS molecule studied in the manuscript. We believe this is also will help to avoid misunderstandings.

Comments 5:

 To summarize, I can not recommend publishing this manuscript in the Materials in its present form.

Response 5:

We believe that our responses above will resolve the concerns of Reviewer, and the revised manuscript could be suitable for Materials Chemistry section of the Molecules journal.

Round 2

Reviewer 2 Report

Comments and Suggestions for Authors

The authors have adequately addressed all the comments in my previous review improving the manuscript. Particularly, they have changed the functional for their DFT calculations and as a result the theoretical outcomes are now in accordance with the experiments. Also, they have provided the necessary reference photoluminescence spectra in both thin film and solution. Indeed, with their work, the authors have showed the potentialities of an extended thiophene-phenylene co-oligomer to implement multi-functional organic optoelectronic devices, combining electroluminesce and photoelectric properties.  Therefore, I can recommend this study of considerable interest to the broad Molecules readership.

Comments on the Quality of English Language

Minor editing is required

Reviewer 3 Report

Comments and Suggestions for Authors

In my opinion, the Authors correctly addressed the reviewers' remarks. The work has been revised to an acceptable level.